# Characterization of Urine-Derived Stromal/Stem Cells from Healthy Dogs and Dogs Affected by Chronic Kidney Disease (CKD)

**DOI:** 10.3390/ani15020242

**Published:** 2025-01-16

**Authors:** Anna Lange-Consiglio, Filippo Tagliasacchi, Fausto Cremonesi, Claudia Gusmara, Claudia Pollera, Paola Scarpa, Giulia Gaspari, Pietro Riccaboni

**Affiliations:** 1Reproduction Laboratory, Department of Veterinary Medicine and Animal Science, Università degli Studi di Milano, 26900 Lodi, Italy; anna.langeconsiglio@unimi.it (A.L.-C.); fausto.cremonesi@unimi.it (F.C.); 2Department of Veterinary Medicine and Animal Science, Università degli Studi di Milano, 26900 Lodi, Italy; filippo.tagliasacchi@unimi.it (F.T.); claudia.gusmara@unimi.it (C.G.); claudia.pollera@unimi.it (C.P.); paola.scarpa@unimi.it (P.S.); pietro.riccaboni@unimi.it (P.R.); 3Laboratorio di Malattie Infettive degli Animali (MiLab), Department of Veterinary Medicine and Animal Science, Università degli Studi di Milano, 26900 Lodi, Italy

**Keywords:** dog, urine, mesenchymal/stem cells, chronic kidney disease, regenerative medicine

## Abstract

Mesenchymal stromal cells are used for their regenerative properties both in human and veterinary medicine. Usually, these cells are collected from bone marrow or adipose tissue, but methods that allow a simple, economical, and non-invasive collection of cells would encourage their use. One easily accessible alternative source would be urine. In 2008, urine mesenchymal stromal cells were isolated for the first time from human urine. These cells come from the kidney and would, therefore, be ideal for use in the treatment of kidney/genitourinary diseases. In this paper, urine mesenchymal stromal cells were isolated from healthy dogs and dogs affected by chronic kidney disease. Three collection methods (spontaneous micturition, bladder catheterization, and cystocentesis) were compared and cells were analyzed for their mesenchymal properties. The isolated cells met the criteria set by the international society of stem cell therapy to be defined as stem cells. Cells isolated from sick dogs proliferated more readily than those isolated from healthy dogs and may provide a therapeutic option for the treatment of chronic kidney disease. However, in this study, cells could only be isolated from urine samples collected through cystocentesis.

## 1. Introduction

Mesenchymal stromal/stem cells (MSCs) can be derived from different tissues but are predominantly sourced from adipose tissue and bone marrow. These two sources have long been studied and have been used in many experimental and preclinical studies. However, these MSCs are obtained through invasive procedures and have prolonged in vitro proliferation times, as studied in equine and canine species [1,2]. Therefore, in recent decades, other sources of MSCs have been studied, such as extra-fetal adnexa in equine species [3] and urine-derived stem cells (USCs) in human medicine [4]. These are considered waste biological material, and cells can be isolated and amplified using simple, non-invasive, painless, and cheap procedures.

The presence of cells with stem cell properties in urine was described in human species by Zhang et al. [4].

Urine-derived stem cells show the classic features of MSCs, expressing some markers related to stemness and pluripotency, high telomerase activity, and do not form teratomas when injected into immunodeficient mice [5]. They have been reported to be able to differentiate into a huge variety of cell types in addition to osteogenic, chondrogenic, and adipogenic lineages [5], as required by the definitions of the international society of stem cell therapy (ISCT) [6]. Indeed, these USCs can differentiate into cardiomyocytes, motor neurons, alveolar epithelial cells, hepatocytes, retinal organoids, renal organoids and, moreover, they can be transformed into induced pluripotent stem cells (iPSCs) [5]. Some authors have described the differentiation of USCs into endothelial, neuronal, skeletal myogenic, renal cells, podocytes, and tubular epithelial cells, which is of great importance considering USC’s potential as renal repair cells [7,8]. Given their characteristics, USCs have been assessed for potential applications in regenerative medicine. They have been used for the first time in urinary tissue engineering [9]: human USC were seeded in bacterial cellulose scaffolds and induced to differentiate into urothelial and smooth muscle cells into athymic mice. In vivo, the cells appeared to differentiate and express urothelial and smooth muscle cell markers. Urine-derived cells have subsequently been employed in other experimental studies to understand their therapeutic potential in various pathologies, as reported in Yu’s review [10]. Stress urinary incontinence was tested on rats; erectile dysfunction was studied in a rat model, in which these cells promote the growth of new blood vessels and nerve cells in the penis, improving erectile function [11]. Also, acute kidney injury was studied [12]: this is an extremely dangerous syndrome in the human species with high morbidity and mortality, which makes it necessary for the wide use of rat models to be uses to test the differentiation, anti-inflammatory, antioxidative stress, and antifibrotic response of human urine-derived cell treatment. Finally, diabetic nephropathy [13] remains one of the most severe complications linked to diabetes mellitus, with damage to glomerular podocytes: in a rat model, the use of human urine-derived exosomal microRNA-16 promoted the proliferation of podocytes and inhibited their apoptosis, alleviating the damage inflicted by diabetic nephropathy and protecting against podocyte injury.

They have also been used in pathologies affecting organs evolutionarily distant from the kidney, such as myocardial cells in a rat model [14], inflammatory bowel disease in a murine model [15], bone and cartilage regeneration in rats [8,16], and wound healing in a mouse model [17].

The potential of USCs for renal/genitourinary repair is particularly attractive as these cells directly originate from this district. Therefore, they might have some specific intrinsic properties that distinguish them from other types of stem cells. According to Kim et al. [18], USCs used in kidney disease could have an enhanced homing effect due to their renal origin; moreover, the same authors demonstrated that these cells express more Klotho protein than other cells. This protein has an important anti-fibrotic effect, and its serum concentration decreases with the progression of chronic kidney disease. Doi et al. [19] demonstrated the reno-protective effects of Klotho on a mouse model of induced renal fibrosis, mediated by the binding of this protein to the TGF-B receptor. Recently, USCs were shown to attenuate renal fibrosis via Klotho activation [20].

The specific origin of USCs and their anti-fibrotic effect, due to the Klotho protein, make these cells an interesting source for kidney regeneration, and thus in the treatment of CKD. Chronic kidney disease is a long-term condition characterized by structural and/or functional kidney abnormalities lasting more than three months. CKD in dogs may occasionally have a familial origin but is more commonly acquired. Potential causes include glomerular diseases, but also infections, nephrotoxicity, previous acute kidney injury (AKI), ischemia, and urinary obstructions. In many cases CKD progresses to interstitial fibrosis which, regardless of the initial cause, determines prognosis.

Since the kidneys have limited regenerative capacity, kidney fibrosis usually proceeds irreversibly leading to the “end stage” condition [21,22].

Potential treatments for this pathology are often aimed at delaying and suppressing fibrosis or inducing the reconstruction of damaged kidney tissue. In a rat model, Zhang et al. [14] demonstrated the protective effect of USCs on the nephron by injecting these cells into the kidney parenchyma, where they showed antioxidant, anti-inflammatory, and antifibrotic activity in CKD. Extracellular vesicles (EVs) present in urine, administered in an experimental murine model of acute kidney injury, also stimulated tubule cell proliferation and reduced inflammation through the transfer of miRNAs and Klotho to resident kidney cells [23].

USCs can be considered as a potential new player for future cell therapy. Urine is a convenient substrate for autologous cell therapy and USCs can be isolated from urine collected through spontaneous micturition, which is a non-invasive and inexpensive method. This suggests the potential for generating a donor biobank without ethical problems.

The literature suggests that the characteristics of USCs could be linked to different isolation techniques or to different donors, as has been shown for other stem cell types [24]. The aim of this study was to compare three different techniques for the collection of urine-derived cells (spontaneous micturition, bladder catheterization and cystocentesis) from healthy dogs and dogs affected by CKD and to compare the proliferative and differentiative potential of these cells.

The results show that it is possible to isolate canine USCs from healthy animals using a variety of urine collection methods, while in CKD dogs this is only possible by cystocentesis. However, this MSC source could represent a useful starting point for future autologous therapy.

## 2. Materials and Methods

### 2.1. Reagent

All reagents were purchased from Sigma-Aldrich (Milan, Italy), while test tubes and culture plates were purchased from Euroclone (Milan, Italy). Blood agar plates were purchased from Microbiol srl (Cagliari, Italy).

### 2.2. Ethics

Urine samples were collected from client-owned dogs referred to the internal medicine unit of the Veterinary Teaching Hospital (VTH) of Università degli Studi di Milano as a part of routine health screening. Written informed consent was provided by the owners for the use of residual aliquots of clinical samples for research purposes. Established internationally recognized ‘best practice’ standards for individual veterinary patient care were followed to collect biological samples. Consequently, according to the guidelines of the institutional Ethics Committee (protocol number 2/2016), no further approval from the Institutional Animal Care and Use Committee was required.

### 2.3. Experimental Design

The study comprised: a selection of animals, the isolation and culture of urine cells, study of their proliferative potential, differentiating capacity, analysis of mesenchymal markers and Klotho expression. All analyzes were performed on pooled urine-derived cells from both healthy and sick dogs.

### 2.4. Animals

Urine samples were collected from nine healthy dogs that were scheduled to undergo ovariohysterectomy or orchiectomy during routine bladder emptying prior to surgery. The samples were collected by either spontaneous micturition, bladder catheterization, or cystocentesis and were transported to the laboratory in sterile syringes or containers without preservatives.

Urine samples from 11 dogs of different breeds, sex, and age affected by CKD were also included in the study. The diagnosis and staging of CKD were made in accordance with the current guidelines of the International Renal Interest Society (IRIS). Each dog met at least one the following criteria: persistent sCr levels above 1.4 mg/dL, and/or persistent renal proteinuria (UPC ratio > 0.5), and/or inadequate urine concentrating ability without identifiable renal cause and/or ultrasonographic abnormalities suggestive of CKD. All the samples were collected by cystocentesis during routine follow-up visits for the monitoring of disease progression.

### 2.5. Bacteriological Analysis

After collection, each sample was immediately transported to the Laboratory of Animal Infectious Diseases (MiLab), University of Milan, Lodi for bacteriological analysis. Briefly, 10 mL of each urine sample was plated onto blood agar plates (5% defibrinated sheep blood (Microbiol, Cagliari, Italy). Plates were incubated aerobically at 37 °C and evaluated after 24 and 48 h. In case of bacterial growth, the isolates were subjected to identification by MALDI-TOF-MS (Matrix-assisted laser desorption ionization-time of flight mass spectrometry) using the Bruker MALDI Biotyper System (Bruker Scientific, Billerica, MA, USA).

### 2.6. Cell Isolation and Culture

The collection of urine by catheterization and cystocentesis was performed in healthy dogs undergoing ovariohysterectomy or orchiectomy after premedication with intramuscular dexmedetomidine 0.004–0.006 mg/kg and butorphanol 0.2–0.4 mg/kg followed by induction with intravenous propofol 2–3 mg/kg with isoflurane 1.0–1.5%. Postoperative pain management consisted of meloxicam 0.2 mg/kg subcutis immediately after recovery from anesthesia followed by meloxicam 0.1 mg/kg orally once daily for 2–4 days.

Catheterization was carried out with sterile catheters.

Cystocentesis samples were collected from three healthy dogs and from all sick dogs using a sterile needle and syringe.

Urine samples were maintained at 4 °C, transported to the laboratory within 2 h, and immediately processed as described by Xu et al. [2] with some modifications, in brief:

Urine samples were transferred into sterile tubes and centrifuged for 10 min at 400× *g* at room temperature (RT). The pellet (not always visible) was gently resuspended with 10 mL phosphate-buffered saline solution (PBS; Euroclone, Milan, Italy) containing 100 U/mL penicillin, 100 μg/mL streptomycin, and 0.25 μg/mL amphotericin B (Euroclone). The mixture was centrifuged at 200× *g* for 10 min, and the supernatant was discarded. A culture medium, “complete urine medium”, i.e., Dulbecco’s Modified Eagle’s Medium high glucose (HG-DMEM, Euroclone) supplemented with 20% of fetal bovine serum (FBS), 10 ng/mL epidermal growth factor (EGF), 100 U/mL penicillin, 100 μg/mL streptomycin, 0.25 μg/mL amphotericin B, 2 mM L-glutamine, 10% platelet rich plasma (PRP, homemade), 1% insulin, transferrin and selenium selenite (ITS), and 10% non-essential amino acids was added to the pellet. Cells detected after centrifugation were counted with Burker’s chamber and stained with Trypan blue 0.4% to detect viable or dead cells. These were cultured in six-well plates in an incubator at 5% CO_2_ and 90% humidity at 38.5 °C at the density of seeding of 10,000 cells/cm^2^. After 48 h, 1 mL of new complete urine culture medium was added to each well followed 48 h later by a 1 mL replacement of fresh complete urine culture medium again. The whole medium was changed after 96 h from the seeding with new culture medium and refreshed every 3 days. The cells were detached by 0.05% trypsin/0.02% EDTA in PBS at 80–90% confluence.

Cells at P0 (when available) and at P1 were studied for morphology. After cryopreservation, at passage 1 and 5 (P1 and P5), pools of cells from healthy and diseased animals were used for the proliferation study (growth curve, doubling time and colony forming unit) and at P3 for differentiation and characterization studies.

### 2.7. Cytological Staining and Immunocytochemical Analysis

At P0, if there was a visible pellet, and at P1, 10,000 cells were plated on a slide placed on the bottom of the well of a six-well polystyrene plate covered with 3 mL of complete urine medium. Cytological analysis was performed after approximately 7 days of culture and after having verified through a microscope that cell proliferation had occurred (by identification of adherence to the slide). These cells were morphologically characterized through the May-Grünwald Giemsa staining and visualized under the microscope at 40× of magnification.

In addition to this staining, the expression of vimentin (with mouse monoclonal, clone Vim V9; Dako, Glostrup, Denmark) and PanCytokeratin (mouse monoclonal, clone A1E; Santa Cruz Biotechnology, Santa Cruz, CA, USA) were analyzed after blocking of endogenous peroxidase by immersion in 3% hydrogen peroxide for 30 min and incubation in normal horse serum diluted 1:10 in PBS. Then, incubation with primary antibodies for 1 h at 37 °C was carried out and primary antibodies were diluted in a blocking solution (anti-vimentin in mouse, m7020, Dako, 1:1000; anti-cytokeratin in mouse, m3515, Dako, 1:1000). After 3 washes in PBS, incubation with biotinylated secondary antibody anti-mouse in horse (Vector) for 30 min was performed, and slides were incubated with ABC (Avidin/Biotin complex, Vector) and then with diaminobenzidine (Vector) for 2 min. At last, slides were contrasted by incubation in hemalum for 2 min, activated by bathing in running water for 5 min and mounted in glycerin for microscope observation.

### 2.8. Cell Proliferation

Each analysis was performed in triplicate.

To obtain cell-proliferation growth curves at P1 and P5, 9 × 10^3^ urine cells were plated into six-well tissue culture polystyrene dishes. Every 2 days, through 13 days of culture, one well of each plate was trypsinized. The total number of live cells was obtained at each time point by staining with the trypan blue dye exclusion method.

Doubling time (DT) of the urine cells, at P1 and P5, was determined by seeding 9 × 10^3^ cells into six-well tissue culture dishes. Cells were trypsinized every 3/4 days, and counted and replated at the same density. The mean doubling time was calculated from day 0 to day 4 for the three replicates. The mean of population doublings (PD) was obtained for each passage according to the formulaCD = log (Nc/No)/log2 and PD = CT/CD
where CD represents cell doubling, Nc represents the number of cells at confluence, No represents seeded cells, and CT represents the culture time [25].

### 2.9. Colony-Forming Unit (CFU) Assays

Colony-forming unit assays were performed to evaluate the clonogenicity of the isolated canine urine-derived cells. At P1, cells were plated at different densities (100, 250, 500, and 1000 cells/cm^2^) in six-well plates and cultured in 5% CO_2_ and 90% humidity at 38.5 °C for 2 weeks in HG-DMEM culture medium enriched with 10% of PRP. Then, colonies were fixed with 4% formalin and stained with 1% methylene blue (Serva, Heidelberg, Germany) in 10 mM borate buffer, pH 8.8 (Fluka BioChemika, Buchs, Swizerland) at room temperature, and washed twice. Colonies formed by 16–20 nucleated cells were counted under a BX71 microscope (Olympus, Tokyo, Japan). The CFU assay was performed independently in each dog in triplicate.

### 2.10. Differentiation Assay

Cells at P3 were seeded at a density of 3 × 10^3^/cm^2^ for all differentiation studies.

Osteogenic differentiation was assessed by incubating cells for up to 3 weeks at 38.5 °C under 5% CO_2_ in modified Romanov et al. medium [26] composed of HG-DMEM medium supplemented with 10% FBS, 100 U/mL penicillin, 100 mg/mL streptomycin, 0.25 mg/mL amphotericin B, 200 mM-glutamine, 10 mM b-glycerophosphate (Sigma), 0.1 mM dexamethasone (Sigma), and 250 mM ascorbic acid (Sigma) [26]. Non-induced control cells were cultured for the same time in standard control medium (HG-DMEM supplemented with 10% FBS, 100 U/mL penicillin, 100 mg/mL streptomycin, 0.25 mg/mL amphotericin B, 200 mM-glutamine). Osteogenesis was assessed by conventional von Kossa staining, using 1% silver nitrate and 5% sodium thiosulphate, which allowed the detection of calcium deposits.

For adipogenic differentiation, near-confluent cells were cultured through three cycles of induction/maintenance to stimulate adipogenic differentiation. Each cycle consisted of feeding the urine-derived cells with supplemented adipogenesis induction medium, followed by culture for 3 days (38.5 °C, 5% CO_2_) and subsequent culture for another 3 days in a supplemented adipogenic maintenance medium. The induction medium consisted of modified Romanov et al. [26] medium, composed of HG-DMEM supplemented with 10% FBS, 100 U/mL penicillin, 100 mg/mL streptomycin, 0.25 mg/mL amphotericin B, 200 mM-glutamine, 10 mg/mL insulin (Sigma), 150 mM indomethacin (Sigma), 1 mM dexamethasone, and 500 mM 3-isobuty-l-methyl-xanthine (Sigma). The maintenance medium consisted of HG-DMEM supplemented with 10% FBS and 10 mg/mL insulin [26]. Noninduced control cells were cultured for the same time in standard control medium. Adipogenesis was assessed using conventional oil red O staining (0.1% in 60% isopropanol) to visualize lipid droplets.

Chondrogenic differentiation was assessed in monolayer culture by incubating cells for 3 weeks in Soncini et al. [27] modified medium, composed of DMEM low-glucose containing 100 nM dexamethasone, 50 mg/mL L-ascorbic acid 2-phosphate, 1 mM sodium pyruvate (BDH Chemicals, Poole, UK), 40 mg/mL proline, ITS (5 mg/mL insulin, 5 mg/mL transferrin, 5 mg/mL sodium selenite; Sigma) and 5 ng/mL TGF-b3 (Peprovet, DBA, Milan, Italy). Noninduced control cells were cultured for the same time in standard control medium. The presence of metachromatic matrix was demonstrated by Alcian blue staining, pH 2.5.

### 2.11. RNA Extraction and RT–PCR Analysis

Expression of specific markers included (CD44, CD29, CD166, CD184), hematopoietic lineage marker (CD34 and CD45), pluripotency markers (OCT*Oct-4* and *Nanog*), and immunogenic antigen (*MHC I* and *MHC II*) were investigated by RT–PCR analysis on undifferentiated cells. Total RNA was extracted at P3 from both kinds of USCs, using TrizolW reagent (Invitrogen, Waltham, MA, USA), followed by DNase treatment according to the manufacturer’s specifications. RNA concentration and purity were measured using a NanoDrop spectrophotometer (NanoDropW ND1000). cDNA was synthesized from 200 ng total RNA, using the iScript retrotranscription kit (Bio-Rad Laboratories, Hercules, CA, USA). Conventional PCR was performed in a 25 mL final volume with DreamTaq DNA Polymerase (Fermentas, St. Leon Rot, Germany). Canine-specific oligonucleotide primers were designed using open source PerlPrimer software v. 1.1.17, based on available NCBI *Canis lupus familiaris* sequences or on mammal multi-aligned sequences. Primers were designed across an exon–exon junction to avoid DNA amplification. Primers were used at 200 nM final concentration and their sequences are shown in Table 1. GAPDH was employed as a reference gene. For differentiation experiments, total RNA was extracted from undifferentiated (control cells) and from induced urine-derived cells, and RT–PCR analysis was performed as described above. To detect the positive osteogenic differentiation, the expression of runt-related transcription factor 2 (*RUNX2*) and bone g-carboxyglutamate protein (*BGLAP*) was evaluated; for chondrogenesis differentiation, collagen type II-a1 (*COL2A1*) and aggrecan (*ACAN*); for adipogenesis differentiation, peroxisome proliferator- activated receptor gamma (*PPARy*) and lipoprotein lipase (*LPL*) were studied, respectively. Primer sequences are listed in Table 1.

### 2.12. Western Blotting for Klotho Protein

Total protein content of urine-derived cells was determined by Bio-Rad Protein Assay (Bio-Rad laboratories, Milano, Italy) and subsequent measurement at 595 nm with Beckman DU 640 spectrophotometer (Beckman, Indianapolis, IN, USA).

For evaluation of Klotho expression, samples containing 40 μg of proteins were mixed with 4× XT Sample Buffer (Bio-Rad laboratories, Milano, Italy), heated for 10 min at 95 °C, and run into parallel lanes into 4–20% CriterionTM TGX Stain-free Precast acrylamide gel (Bio-Rad laboratories, Milano, Italy) for SDSPAGE together with a molecular weight size marker (prestained dual color Protein Ladder; (Bio-Rad laboratories, Milano, Italy). For Western blotting analysis, proteins were electro-transferred by Trans Blot SD semi-dry apparatus (Bio-Rad laboratories, Milano, Italy) to an immobilon-P membrane (Millipore, Bedford, MA, USA) and stained with Coomassie blue to verify the efficacy of the transfer. The membrane was cut into two pieces: one was hybridized with 1 µg of mouse monoclonal primary antibody against Klotho protein (sc-515942 Santa Cruz Biotechnology, Inc., Dallas, TX, USA) and the other with 1 µg of anti GAPDH (sc-47724 Santa Cruz Biotechnology, Inc., Dallas, TX, USA) by the SNAP i.d. Protein Detection System (Millipore, Bedford, MA, USA). The hybridization signal was evidenced by the Vectastain elite system (Vector-Laboratories, Burlingame, CA, USA) that uses a biotinylated universal antibody, the ABC reagent, and the diaminobenzidine (DAB) peroxidase substrate solution, according to the procedure suggested by the company. In the negative control test, the hybridization procedure was performed omitting the primary antibodies. Membranes were imaged with a Coolpix P5100 (Nikon, Osaka, Japan). The optical density of each protein band was quantified using Quantity one 1-D Analysis software (BioRad, Milano, Italy). The values expressed as arbitrary units (A.U.) are presented as the ratio of the specific protein to the corresponding GAPDH optical density.

### 2.13. Statistical Analysis

Statistical analysis was performed using GraphPad Instat 3.00 for Windows (GraphPad Software, La Jolla, CA, USA). Three replicates for each experiment (growth curve, DT and CFU) were performed and the results are reported as mean ± standard deviation (SD).

One-way analysis of variance (ANOVA) for multiple comparisons by Student–Newman–Keuls multiple comparison tests was used. CFU comparison among different cell plating densities inside each group and between groups of the same cell density were analyzed. *p* < 0.05 was considered as significant.

## 3. Results

### 3.1. Animals

Table 2 shows the characteristics of each healthy dog. Table 3 shows the characteristics of each sick animal.

### 3.2. Bacteriological Results

All urine samples were sterile after bacteriological analysis.

### 3.3. Isolation of Urine-Derived Cells

The average number of cells collected from each technique of urine collection in healthy and sick dogs is shown in Table 4.

In healthy dogs, cells were always obtained through all collection methods.

In sick dogs, only cystocentesis was performed, and cells were always found.

### 3.4. Cytological Staining and Immunocytochemical Analysis

At passage 0 (P0), urine-derived cells from both sick and healthy dogs can assume intermediate phenotypes with less elongated and more polygonal shapes (Figure 1A). At passage 1, cells showed an elongated, fibroblast-like shape (Figure 1B).

At P0, immunocytochemical characterization highlighted the co-existence of mesenchymal and epithelial expression markers (vimentin and cytokeratin, respectively), as shown in Figure 2.

### 3.5. Cell Proliferation Analysis

#### 3.5.1. Healthy Dogs

Urine-derived cells obtained by cystocentesis (animals B, E and G) reached a confluence of 80% after 15 days of culture at P0, and this primary culture was expanded and studied until P5.

Cells obtained by bladder catheterization and spontaneous micturition reached the same confluence, but after 25 days and after the first passage it was no longer possible to detach them, despite the use of trypsin or scrapers; therefore, they could not be characterized (Table 5).

Urine-derived cells showed a growth curve with a lag phase of 48 h and a subsequent log phase between 4 and 9 days at P1, while the growth curve always remained in lag phase at P5 (Figure 3). The DT for urine-derived cells was constant up to P3, then decreased statistically significantly (*p* < 0.05) at P4. The mean DT value was 2.04 ± 0.19 days (Figure 4).

#### 3.5.2. Sick Dogs

Urine was collected from 11 CKD dogs, but cells were isolated and expanded from only three of these. These cells reached a confluence of 80% after 8 days of culture at P0 and were expanded and studied until P5.

Urine-derived cells at P1 and P5 demonstrated a growth curve with an initial lag phase of 48 h and subsequent log phase that was more intense at P1. At P5, these cells showed a slight minor proliferation compared to those at P1 (Figure 5).

In CKD dogs, the DT for urine-derived cells decreased in a statistically significant manner (*p* < 0.05) until P3, then remained constant until P5. The mean DT value was 1.7 ± 0.2 days (Figure 6).

The mean DT in sick dogs was lower than that of healthy dogs and the difference was statistically significant with a *p* value of 0.0219. This result highlights a greater proliferative capacity of the urine-derived cells from CKD dogs.

### 3.6. CFU Assays

The number of cell colonies formed was counted at P1 after seeding cells at different densities/cm^2^. For each cell population (healthy and sick derived cells), there was a statistically significant increase in CFU frequency with increasing cell seeding densities for both kinds of samples (Table 6 and Figure 7). There were no statistical differences between the number of CFUs between healthy and sick urine-derived cells.

### 3.7. In Vitro Differentiation

The multi-differentiative potential of urine-derived cells in healthy and CKD dogs was evaluated at P3. After positive results of osteogenic, adipogenic, and chondrogenic differentiation, urine-derived cells are called “urine mesenchymal stromal/stem cells (USCs)”.

#### 3.7.1. Osteogenic Differentiation

After 21 days of induction, the osteogenic differentiation of urine-derived cells from healthy and sick dogs was confirmed by von Kossa stain, which highlighted calcium deposits. The cells changed their morphology. Microscopic images show no apparent differences in staining intensity between healthy and sick dogs. The control was negative in staining, showing no mineralized matrix. Analysis of the expression of osteogenic markers *BGLAP* and *OPN* by RT-PCR confirmed osteogenic induction (Figure 8A,B and Figure 9A,B).

#### 3.7.2. Adipogenic Differentiation

Urine-derived cells from both healthy and sick dogs showed the ability to differentiate into the adipogenic lineage, as demonstrated by the positive result of Red Oil O staining after 3 weeks of culture in the adipogenic medium. Microscopic images show no apparent differences in staining intensity between healthy and sick dogs. Cells maintained in the standard medium showed no lipid deposits and, therefore, staining was negative. The cells induced to differentiate revealed an increased expression of *PPAR-γ* and *adiponectin* compared to the control that was negative (Figure 8C,D and Figure 9C,D).

#### 3.7.3. Chondrogenic Differentiation

Urine-derived cells derived both from healthy and sick dogs had the ability to differentiate into the chondrogenic lineage, as reflected by Alcian Blue staining. Urine-derived cells derived from sick dogs exhibited more intense staining than those from healthy dogs. The control failed to stain as expected. Analysis of the expression of the chondrogenic markers, *COL2A1* and *ACAN* by RT-PCR confirmed the induction compared to the control, which did not show expression of the markers (Figure 8E,F and Figure 9E,F).

### 3.8. RNA Extraction and RT-PCR Analysis

To characterize urine-derived cells, a multi-step RT-PCR was set up. All cells from healthy and sick dogs expressed MSCs-specific markers (*CD166*, *CD117*, *CD29*, *CD90*, *CD73*), embryonic marker (*Nanog*, *Oct4*, *Sox2*), hematopoietic marker (*CD34*), histocompatibility complex *MHC I* but not *MHC II* marker (Figure 10).

*GAPDH* was used as a housekeeping gene.

### 3.9. Western Blotting for Klotho Protein

Klotho protein expression was confirmed by Western blotting (Figure 11).

## 4. Discussion

Cell therapy could be a useful treatment for genito-urinary pathologies in domestic animals. Since Italian law requires autologous treatments, USCs have ethical advantages over other MSCs, as they can be repeatedly obtained from urine without causing any pain or side effects in the patients [28]. However, the isolation of urine-derived cells is only apparently simple: these cells can survive only for a few hours in urine, that indeed represent a hostile environment for cell survival due to lack of nutrients, presence of toxic metabolic waste, high osmotic pressure, and non-physiological pH value. This toxic environment alters the cell membrane and causes cell lysis. Different strategies can be used for their preservation: the addition of a preservation medium or the storage of urine samples at 4 °C to maintain cell membrane stability by slowing metabolism and preventing cell lysis [29].

In our study, it appeared to be a correlation between donor age and the number of cells cultured. Studies have shown that cells isolated from young children (non-newborns) have better proliferation and differentiation abilities [30]. The clinical condition of the subject also influences the success of isolation of these cells. For example, cells obtained from diabetic donors have never been successfully isolated and cultured [4,31,32]. There are no studies reporting the isolation of urine-derived cells from dogs affected by kidney diseases. Our study evaluated the biological characteristics of USCs isolated from different urine collection methods in healthy dogs. We compared these cells with USCs isolated from CKD affected dogs, to understand if they show the same characteristics. Indeed, it would be advantageous to perform autologous cell therapy and prevent immune rejection risk in dogs with CKD.

The isolation of urine-derived cells from healthy dogs was carried out in samples from spontaneous micturition, bladder catheterization, and cystocentesis. The isolation in CKD dogs was performed only by cystocentesis.

Urine samples were stored at 4 °C immediately after collection and processed within 2 h from their arrival in the laboratory to minimize the exposure of the cells to the toxic environment of urine. In healthy dogs, cells were isolated from all samples (regardless of the collection technique), but those obtained by spontaneous micturition and bladder catheterization underwent senescence at P1, making it impossible to proceed with the study. Thus, in healthy dogs only cells obtained by cystocentesis were studied. In CKD dogs, cells were isolated only from three of eleven animals belonging to this group and, even in these cases, the urine collection was performed by cystocentesis.

We cannot provide explanations for these problems, as the only study concerning the isolation of USCs from dog species [2] was performed on healthy beagle dogs, where urine was collected by a sterile catheter, and the sex and age of these animals were not reported. In our study, no cells were able to proliferate when obtained by bladder catheterization or spontaneous micturition.

We have analyzed different parameters to understand this aspect; for example, the age of the animals was not an influencing factor for the culturing of these cells. Indeed, cells were easily obtained by cystocentesis from older dogs (including a 12-year-old) and were able to proliferate.

Another critical issue encountered in our study was the isolation of urine-derived cells only in three of the 11 CKD dogs from which urine was collected through cystocentesis. We did not find any correlation with age, breed, severity of the disease and urinary parameters but only a correlation with sex. Indeed, all three of these dogs were spayed females. Comparably, also among the healthy dogs, urine-derived cells were successfully isolated and cultured only from three young females (animal B, E, and G of 1–2 years old). In the literature, there are no data about the outcome of isolation of USCs correlated with the sex of the donor.

Initially, to try to culture all kinds of cells (from spontaneous micturition, bladder catheterization, and cystocentesis), a basic culture medium with HG-DMEM, 20% of fetal calf serum (FCS), and antibiotic and antimycotic, was enriched by 1% of ITS, or 1% of non-essential amino acid, or 2% of renal epithelial cell growth medium (Lonza, Milan, Italy), as described in dog species by Xu et al. [2], or 10 ng/mL of EGF, adding one supplement at a time. Despite this, cells from spontaneous micturition and bladder catheterization died compared to cells obtained from cystocentesis.

The idea of adding 10% of platelet-rich plasma (PRP) to the culture medium after a dose response curve was suggested to promote the survival of these cells. Indeed, PRP is a source of cytokines, chemokines, and growth factors involved in stimulating cell proliferation, inducing tissue regeneration. This product is used by our research team for in vitro and in vivo studies [33,34,35,36,37]. Then, the final complete urine medium used in our study was composed by HG-DMEM enriched of 20% of FCS, 100 U/mL penicillin, 100 μg/mL streptomycin, 10 ng/mL epidermal growth factor, 0.25 μg/mL amphotericin B, 2 mM L-glutamine, 1% insulin, transferrin and selenium, 10% non-essential amino acids and 10% PRP (homemade). This medium did not alter the growth curve of cells obtained by cystocentesis in healthy or sick dogs but did not promote the survival of cells obtained by spontaneous micturition and bladder catheterization, which were therefore abandoned in our experiment.

Cells obtained by cystocentesis in the three CKD dogs, and by cystocentesis in the three healthy dogs were studied for cell proliferation, differentiation, and marker expression analyzes.

Studies on the proliferative capacity of urine-derived cells have shown a greater proliferative capacity of these cells in diseased animals compared to healthy animals. In sick dogs, the confluence is reached in about 7–8 days (compared to 15 days in healthy dogs) and the mean DT is statistically lower than that of healthy dogs. These data are not in agreement with those obtained in human studies by Guan et al. and Chen et al. [8,17], who reported 80% of confluence reached in 3 and 5 days. Also with canine USCs, it was reported that cells reached 70–80% of confluency after 5 days [2]. Instead, the DT obtained in sick dogs was similar to that obtained in human USCs from Zhang et al. [4]. Of course, there are many variables related to the different laboratory procedures that can explain these differences of a few days.

At passage 1, all isolated cells began to show a fibroblast-like morphology, typical of mesenchymal stem/stromal cells and, indeed, the isolated cells (both in healthy and sick dogs) showed expression of MSC markers (*CD166*, *CD117*, *CD29*, *CD90* and *CD73*) and embryonic markers (*Nanog*, *Oct-4* and *Sox*). Furthermore, all cells expressed *MHC I* at passage 1 and 5, but not *MHC II*. This suggests the possible therapeutic use of these cells not only for autologous therapy but also for allogenic treatment. Furthermore, urine-derived cells express Klotho protein [18], but in sick dogs this expression was significantly greater compared to healthy dogs. This result is very interesting considering that the use of USCs, and hence Klotho protein, in experimental models of CKD has been shown to have a protective effect on the nephron. At renal level, Klotho protein blocks various signaling pathways that lead to the development and maintenance of renal fibrosis after kidney damage in a mouse model [18].

The analysis that confirmed the stem cell property of isolated cells (making characterization of USCs) was the differentiation study. All cells demonstrated osteogenic, adipogenic, and chondrogenic differentiation capacities. Data deriving from the literature suggest their differentiation into many other cell types, in addition to those described [5].

The morphology, phenotype, proliferation and multipotency of isolated cells in our study is consistent with currently accepted biological characteristics of MSCs.

The focus of regenerative medicine is the possibility to collect cells in a noninvasive way, and the availability of highly proliferative cells that can differentiate. Comparison of proliferation data of MSCs is difficult among different laboratories. In our experience, we worked with equine cells isolated from bone marrow and amniotic membrane and we highlighted a notable proliferative difference between the two cell lines [38]. The average DT of amniotic MSCs was 2 days shorter than that observed in bone marrow MSCs (1.17 vs. 3.27 days), confirming the data of Guest et al. [39] on bone marrow derived cells. In addition, as extra-fetal mesenchymal/stromal cells, USCs had the ability to give rise in vitro to clones with frequency that increased with cell-seeding densities up to 1000 cells/cm^2^, indicating some paracrine signaling between urine-derived cells, which may potentiate CFU formation during cell culture [40,41].

More satisfactory results have been achieved in vitro with mature adipose tissue-derived cells capable of differentiation into several mesenchymal derivatives [42], but the collection of bone marrow and adipose tissue is still invasive. Stromal cells from urine could overcome these limitations and potentially offer new approaches in regenerative medicine for autologous cell therapy. Indeed, USCs were present and were isolated, with minimal differences, both in healthy and in sick dogs. Despite their successful isolation resulting only in a small percentage of dogs, USCs could represent a valid starting point for the future autologous therapy. CKD prevalence varies widely, affecting 0.05–3.74% of the general dog population [43,44]. However, prevalence varies across different geographical areas. In fact, more recent observations evidenced the high prevalence of proteinuria in dogs of the Mediterranean area due to the spread of infectious diseases that often cause immune-complex glomerulonephritis [45,46]. The results of this study, in the context of a degenerative and irreversible disease, lay the foundation for promising applications of USCs in the field of canine regenerative medicine.

## 5. Conclusions

Deriving from a convenient source and being theoretically very simple to obtain, canine USCs are considered very promising. However, issues remain with cell isolation and culture. It is very important to preserve the viability of the few cells present in the samples immediately after collection by addition of supplements such glucose, serum, and other sources of nutrition. Despite the difficulty in working with these cells, they could be considered a new source to be used in regenerative medicine for the treatment of CKD in dogs.

## Figures and Tables

**Figure 1 animals-15-00242-f001:**
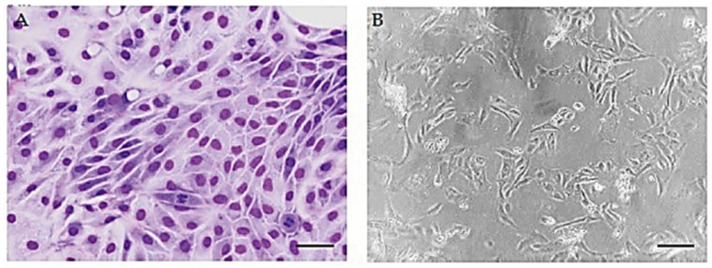
Morphology of urine cells. (**A**) May-Grünwald Giemsa with different cellular morphology at P0. Scale bar 15 µm. (**B**) Fibroblastoid-like morphology of urine-derived cells at P1. Scale bar 10 µm.

**Figure 2 animals-15-00242-f002:**
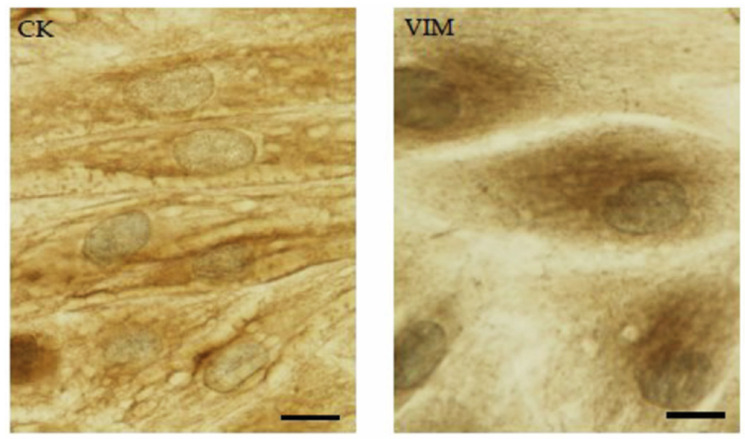
Cytokeratin (CK; scale bar = 20 µm) and Vimentin (VIM; scale bar 15 µm) expression.

**Figure 3 animals-15-00242-f003:**
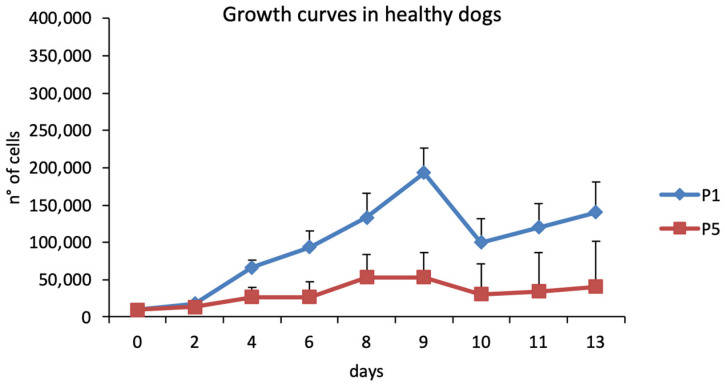
Growth curve in healthy dogs.

**Figure 4 animals-15-00242-f004:**
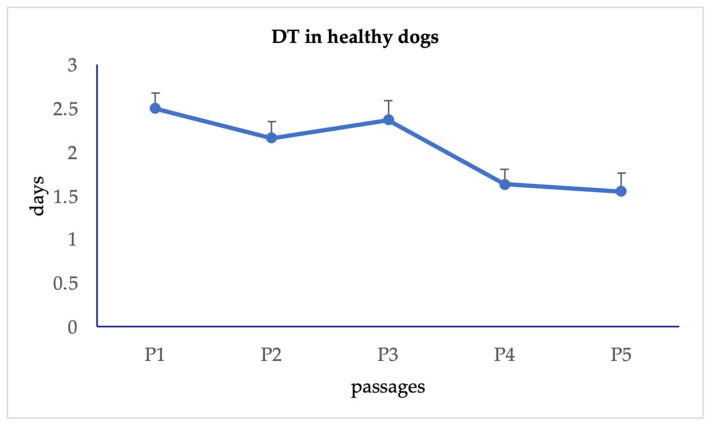
Doubling time in healthy dogs.

**Figure 5 animals-15-00242-f005:**
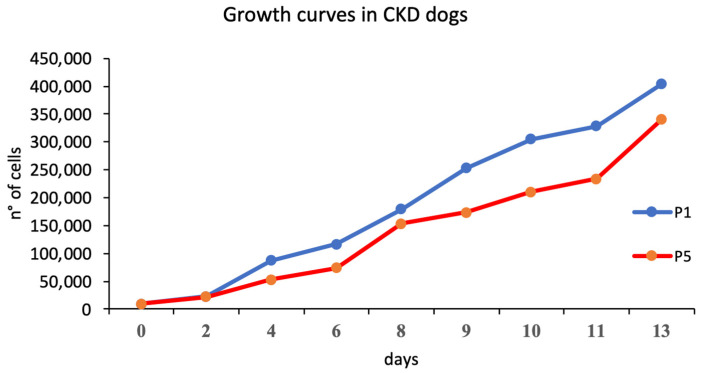
Growth curve in CKD dogs.

**Figure 6 animals-15-00242-f006:**
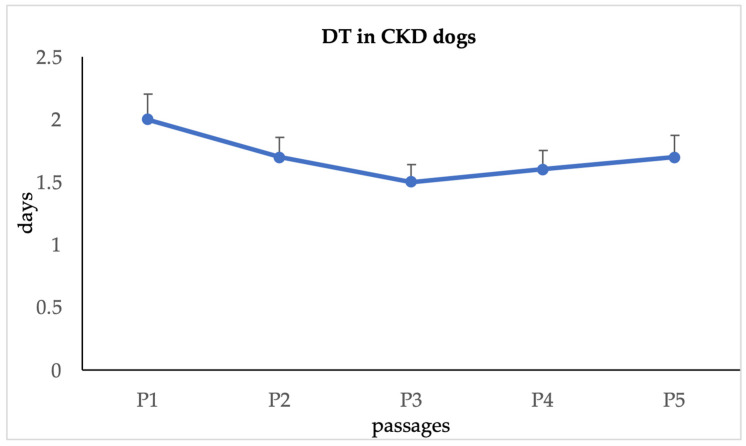
Doubling time in sick dogs.

**Figure 7 animals-15-00242-f007:**
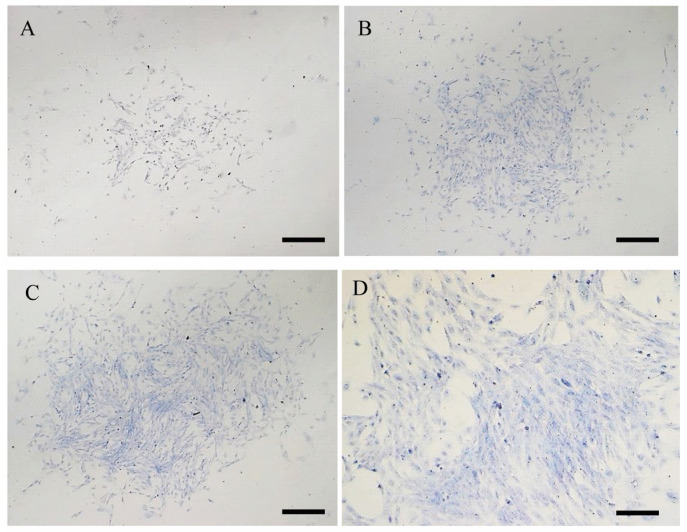
Colony forming unit obtained with different seeding density: (**A**) 100 cells/cm^2^; (**B**) 250 cells/cm^2^; (**C**) 500 cells/cm^2^; (**D**) 1000 cells/cm^2^. Magnification 20×. Scale bar 10 µm.

**Figure 8 animals-15-00242-f008:**
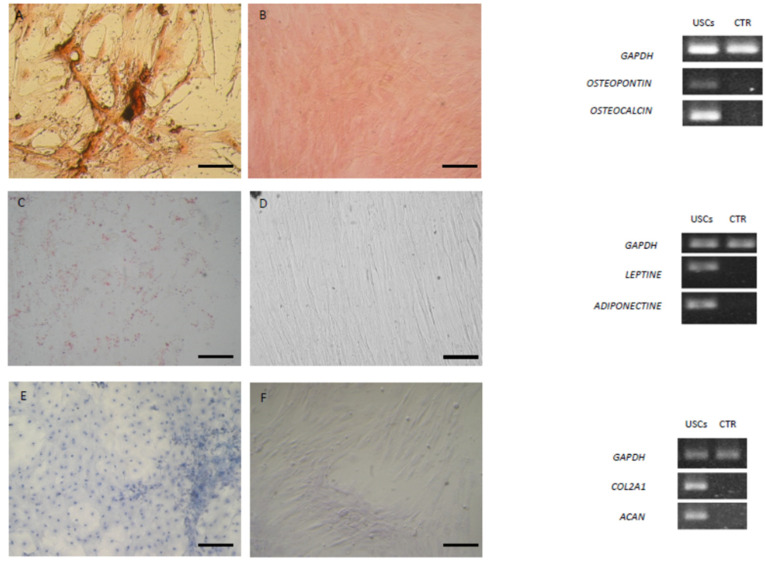
Staining of differentiated and control undifferentiated USCs derived from healthy dogs and respective molecular expression. (**A**,**B**) von Kossa staining after osteogenic induction and RT-PCR analysis of osteopontin (*OPN*) and *osteocalcin* (*BGLAP)*. (**C**,**D**) Oil red O cytoplasmic neutral lipids after adipogenic induction and RT-PCR of *leptin* and *adiponectin*. (**E**,**F**) Alcian blue staining after chondrogenic induction and RT-PCR of collagenase (*COL2A1*) and aggrecan (*ACAN*). Magnification 20×; scale bar = 20 µm. *GAPDH* was employed as a reference gene.

**Figure 9 animals-15-00242-f009:**
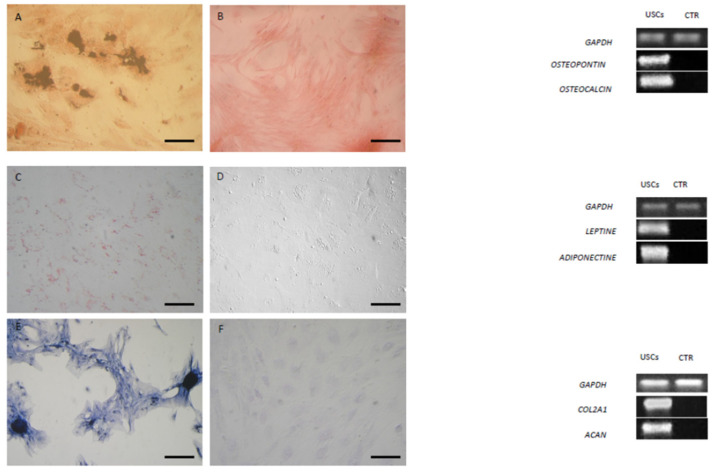
Staining of differentiated and control undifferentiated USCs derived from sick dogs and respective molecular expression. (**A**,**B**) von Kossa staining after osteogenic induction and RT-PCR analysis of osteopontin (*OPN*) and osteocalcin (*BGLAP)*. (**C**,**D**) Oil red O cytoplasmic neutral lipids after adipogenic induction and RT-PCR of *leptin* and *adiponectin*. (**E**,**F**) Alcian blue staining after chondrogenic induction and RT-PCR of collagenase (*COL2A1*) and aggrecan (*ACAN*) Magnification 20×; scale bar = 20 µm. *GAPDH* was employed as reference gene.

**Figure 10 animals-15-00242-f010:**
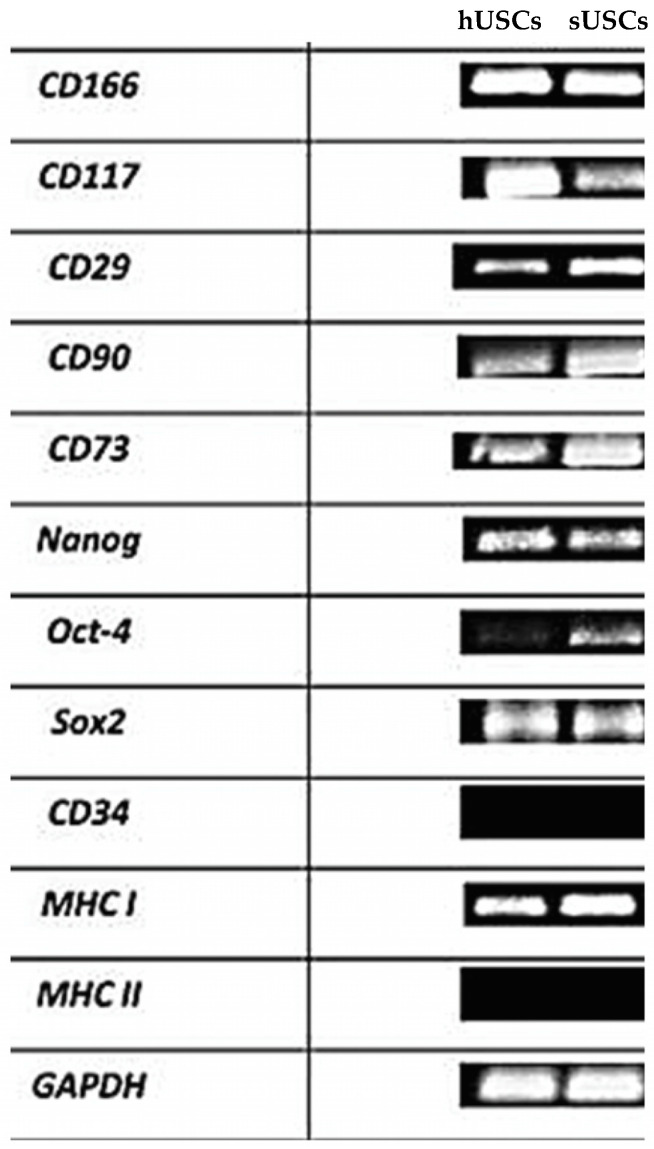
RT-PCR analysis of mesenchymal, embryonic, hematopoietic and histocompatibility complex gene expression on USCs of healthy (hUSCs) and sick (sUSCs) dogs. *GAPDH* was used as reference gene.

**Figure 11 animals-15-00242-f011:**
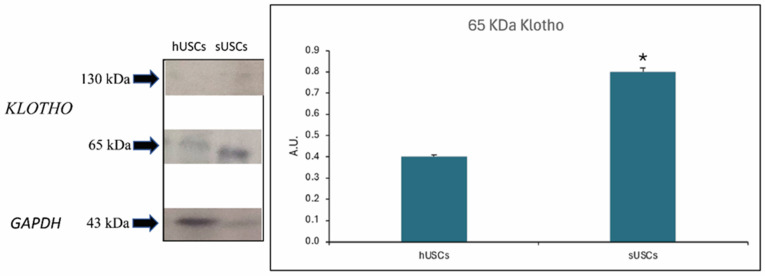
Klotho protein expression was confirmed by Western blotting. Results are shown as the mean  ±  SEM. * *p* < 0.05. Legend: hUSCs = USCs from healthy dogs; sUSCs: = USCs from sick dogs.

**Table 1 animals-15-00242-t001:** Primer sequences used for qRT-PCR analysis.

Markers	Sequence (5′ → 3′)	Product Size (bp)	Annealing Temperature
Glyceraldehyde-3-phosphate dehydrogenase (*GAPDH*)	F: GCAAAGTGGACATTGTCGCCATC R: AGCTTCCCATTCTCAGCCTTGACT	124 bp	64.4 °C
CD34 molecule (*CD34*)	F: ACCAGAGCTACTCCCGAAAG R: TAAGGGTCTTCGCCCAGC	139 bp	59 °C
Integrin β-1 (*CD29*)	F: TAAGAGTGCCGTGACAACCG R: TTCAGAACCTGCCCATAGCG	154 bp	60 °C
CD73 enzyme (*CD73*)	F: ATTCGAGCAAGTGCGTCAAC R: TCGTAACCCAAGGCGTTCAT	193 bp	59.5 °C
Thy-1 Antigen (*CD90*)	F: GCTAACAGTCTTGCAGGTGG R: AGAAGTTGGTTCGAGAGCGG	212 bp	59.5 °C
Tyrosine-protein kinase Kit (*CD117*)	F: GGACCGAAGGAGGCACTTAC R: AACGGAACATCTCTGCTCGG	206 bp	60 °C
ALCAM (*CD166*)	F: TGGTCACAGAGGACAACGT R: CCACGTGATGTTGCCATCTG	167 bp	59.5 °C
Endoglin (*CD105*)	F: AGTTCTCCCGAAGCCTGGTC R: GTGCGAGTGGATGTACCAGAG	104 bp	61 °C
Major histocompatibility complex I (*MHC I)*	F: TGGAGAGGAGCAGAGCTACAC R: CTGTCACTGCCTGCAGCCT	225 bp	61 °C
SLA-DRA1 (*MHC II*)	F: TCTACACCTGCCAAGTG R: CCACCATGCCCTTTCTG	178 bp	55 °C
Transcription factor Oct-4 (*Oct4*)	F: GTTCAGCCAAACGACCATCTG R: TCTCTGCCTTGCATATCTCCTG	140 bp	59.8 °C
Transcription factor Nanog (*Nanog*)	F: AACTTCACCAATGCCTGAG R: CTGATCTTCTGCTTCTTGACTG	234 bp	56 °C
Osteocalcin (*BGLAP*)	F: TCAACCCCGACTGCGACG R: TTGGAGCAGCTGGGATGATGG	204 bp	62.4 °C
Osteopontin (*OPN*)	F: TTGCTAAAGCCTGACCCATCT R: CGTCGTCCACATCGTCTGT	145 bp	59.4 °C
Leptin	F: AGCCTTTCGACCATCAAGCA R: CAACTTGTGTTGCGTGGGAG	100 bp	59.9 °C
Adiponectin (*ADIPQ*)	F: TATGATGTCACCACTGGCAAATT R: TAGAGGAGCACAGAGCCAGAG	185 bp	59 °C
Collagen type 2 alpha 1 (*COL2A1*)	F: ATCGAGATCGCCACCTACAG R: CAGGCTGGTTTCTCGGATCT	102 bp	59 °C
Aggrecan (*ACAN*)	F: CAGGAGAAACAGGGCCTACA R: GCTCCAACTTAGGGTCCAAGA	193 bp	59 °C

**Table 2 animals-15-00242-t002:** Healthy dogs: signalment and urine collection method.

Sample Number	Breed	Gender	Age	Collection Method
A	German Shepherd	M	7 years	Bladder catheterization
B	Pitbull terrier	F	2 years	Cystocentesis
C	Labrador retriever	M	1 year	Bladder catheterization
D	Labrador retriever	M	5 years	Spontaneous micturition
E	Border Collie	F	2 years	Cystocentesis
F	Belgian Shepherd Malinois	M	6 months	Spontaneous micturition
G	Labrador retriever	F	1 year	Cystocentesis
H	Labrador retriever	F	1 year	Bladder catheterization
I	Mixed Breed	F	4 years	Spontaneous micturition

**Table 3 animals-15-00242-t003:** CKD dogs, signalment, urinalysis, and IRIS staging.

Sample Number	Breed	Gender	Age	IRIS Staging	Collection Method	Notes
1	Mixed breed	M	5 years and 7 months	Stage 1	Cystocentesis	
2	English bulldog	FS	1 year and 2 months	Stage 3	Cystocentesis	
3	German hound	MN	7 years and 7 months	Stage 3	Cystocentesis	
4	Mixed breed	M	15 years	Stage 2	Cystocentesis	Suspected HC
5	Mixed breed	FS	8 years and 9 months	Stage 3	Cystocentesis	
6	Nova scotia duck tolling retriever	F	1 year and 8 months	Stage 1	Cystocentesis	
7	Epagenul Breton	FS	8 years	Stage 1	Cystocentesis	Microhematuria; amyloidosis; leishmaniosis
8	Mixed breed	FS	15 years and 8 months	Stage 1	Cystocentesis	HC Dead
9	Dachshund	M	9 years	Stage 1	Cystocentesis	Crystalluria
10	Mixed breed	FS	10 years and 4 months	Stage 1	Cystocentesis	Diabetes
11	Dogue de Bordeaux	FS	9 years and 8 months	Stage 1	Cystocentesis	UTI

Legend; HC = hyperadrenocorticism; UTI = urinary tract infection.

**Table 4 animals-15-00242-t004:** Method of collection and number of cells collected.

Healthy Dogs			Sick Dogs		
N° Samples	Collection Method	N° Cells	N° Samples	Collection Method	N° Cells
3	Spontaneous micturition	57,000 ± 8300	-	-	-
3	Bladder catheterization	33,000 ± 2500	-	-	-
3	Cystocentesis	93,000 ± 11,327	11	Cystocentesis	122,500 ± 10,584

**Table 5 animals-15-00242-t005:** Number of cells at P1 and P2 in healthy dogs.

Method of Collection	N° Cells at P1	N° Cells at P2
Spontaneous micturition	10,000 ± 840	No cells
Bladder catheterization	10,000 ± 1023	No cells
Cystocentesis	250,000 ± 12,538	300,000 ± 17,024

**Table 6 animals-15-00242-t006:** CFU assay in healthy urine-derived cells.

Density Cells/cm^2^	Total Cells	CFU	1 CFU Each	CFU	1 CFU Each
		Healthy Dogs		Sick Dogs	
100 cells/cm^2^	950	20 ± 1.76 ^a^	47.5	22 ± 1.9 ^a^	43.18
250 cells/cm^2^	2375	30 ± 2.54 ^b^	79.16	35 ± 2.84 ^b^	67.86
500 cells/cm^2^	4750	60 ± 3.18 ^c^	79.16	57 ± 2.31 ^c^	83.33
1000 cells/cm^2^	9500	82 ± 5.82 ^d^	115.85	79 ± 3.73 ^d^	120.25

Different small letters superscripts (^a^, ^b^, ^c^, ^d^) indicate statistically different comparison (*p* < 0.05) between cell density.

## Data Availability

There is no new data than that presented in this manuscript.

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
