# Peer review of "Characterization of Urine-Derived Stromal/Stem Cells from Healthy Dogs and Dogs Affected by Chronic Kidney Disease (CKD)"

_animals, 2025, doi:10.3390/ani15020242_

Round 1
Reviewer 1 Report
Comments and Suggestions for Authors
The study describes the possibility of isolating mesenchymal stromal cells from urine samples collected from dogs with and without renal failure. Given the collection method and the substrate under investigation, it is necessary for the authors, in the introduction, to provide a more detailed explanation, supported by references, of the potential advantages and the specific pathologies these cells could be used for. Similarly, the discussion should include this information, comparing the results obtained with those reported in the literature for MSCs isolated from other sources.
In the discussion section, the authors refer to culture experiments involving the cells studied. Why are these data not reported in the results of the current work? To substantiate the points discussed, it is necessary for the authors to expand the materials and methods and results sections to include the omitted data.
In the results section, it is necessary to include the authors' observations regarding also osteogenic and adipogenic differentiation. For adipogenic differentiation, the authors could reference the semi-quantitative scoring system used in the evaluation of adipogenic differentiation of mesenchymal stromal cells by Burk et al. (2013).
The quality of the images is poor. The authors need to improve the quality of the images provided.
Author Response
The Authors thank the Reviewer for the considerations and helpful suggestions. According to the comments and suggestions, we have carefully evaluated all critical points and the manuscript has been thoroughly revised. The Authors hope that now the manuscript is suitable for publication on “Animals”.
The study describes the possibility of isolating mesenchymal stromal cells from urine samples collected from dogs with and without renal failure. Given the collection method and the substrate under investigation, it is necessary for the authors, in the introduction, to provide a more detailed explanation, supported by references, of the potential advantages and the specific pathologies these cells could be used for. Similarly, the discussion should include this information, comparing the results obtained with those reported in the literature for MSCs isolated from other sources.
ANSWER: the authors thank the referee for her/his suggestions, which helped to improve the manuscript.
The authors have added different sentences in Introduction and Discussion sessions, as suggested from you.
In the discussion section, the authors refer to culture experiments involving the cells studied. Why are these data not reported in the results of the current work? To substantiate the points discussed, it is necessary for the authors to expand the materials and methods and results sections to include the omitted data.
ANSWER: the authors thank the referee for her/his suggestions, which helped to improve the manuscript. The development of the protocol for the culture of cells isolated from urine was very long and laborious. Although there are papers in the human and canine field regarding cell culture media, in our laboratory we were unable to cultivate these cells for long time. It took two years of attempts to succeed in culturing them, generating such an abundance of data, that we are thinking of writing a paper to fine-tune the culture protocol of cells isolated from urine, describing in detail also the production of platelet-rich plasma and the dose-response curves that led us to choose the percentage of 10% PRP. We believe that adding all this data to fine-tune the protocol in this paper would distort the objective of the paper itself.
In the results section, it is necessary to include the authors' observations regarding also osteogenic and adipogenic differentiation. for adipogenic differentiation, the authors could reference the semi-quantitative scoring system used in the evaluation of adipogenic differentiation of mesenchymal stromal cells by burk et al. (2013).
ANSWER: the authors thank the referee for her/his suggestions, but we have not an automatic microscopic system to detect and study the microscopic photographies. The authors had only the possibility to study the different microscopic images observing a greater intensity of staining in the chondrogenic differentiation of cells obtained from diseased individuals compared to healthy ones. No apparent difference has been detected in the microscopic images for the osteogenic and adipogenic differentiation between healthy animals and diseased animals. In any case, a sentence has been added in the results for osteogenic and adipogenic differentiation highlighting the apparent lack of difference in the differentiative capacity of cells isolated from healthy and sick dogs.
The quality of the images is poor. The authors need to improve the quality of the images provided.
ANSWER: the authors thank the referee for her/his suggestions. The authors have been trying to improve the images.
We feel that we have addressed all of the queries raised by the referees and hope that the paper is now acceptable for publication in Animals.
We thank you in advance for your time and consideration.
On behalf of all authors best regards,
Giulia Gaspari
Reviewer 2 Report
Comments and Suggestions for Authors
Dear editor,
The article “Characterization of urine-derived stromal/stem cells from healthy dogs and dogs affected by chronic kidney disease (CKD)” describes properties of USCs derived from dogs using three experimental approaches.
The introduction section lacks important information about the species of the described cells. For each article, the source shall be presented: human, canine, murine (some other?).
Figures 9 and 10 in the additional file are of bad quality.
The idea that “It is very important to preserve the viability of the few cells present in the samples, immediately after collection by addition of supplements such glucose, serum and other sources of nutrition, as PRP.” was not shown in the current study and thus shall not be in the Conclusion section.
There are several minor issues listed below. Also extensive correction of English language is necessary.
Overall, I recommend minor revision of the MS.
L39 What is “immunogenic antigen”? Most antigens are immunogenic by definition.
L56-L115 For each article, the source shall be presented: human, canine, murine (some other?)
L113-114 “The results show that it is possible to isolate canine USCs from both healthy and CKD dogs, using a variety of urine collection methods.” - That is not true, only one method was successful, cells obtained by different methods were not tested.
L174 “1% antibiotic/antimycotic” – which one?
L249-251 Why two media are used, induction and maintenance? A reference to this protocol shall be included.
L269 here and please check throughout the text: if RNA of DNA is analyzed, italics shall be used, if protein – not italics. Also “Oct4” is preferable here as capital letters are used only for human genes, not dogs.
L276 25 microliters?
L281 “T1 GAPDH” – What is “T1”?
L297 Celsius degree sign is wrong.
L365 (Figure 3) and all graphs: why whiskers are looking both up only and up and down on one graph? Why on other graphs they are looking only up? Please use only one option on all the graphs.
L376 “At P5, these cells showed a slight minor proliferation compared to those at P1” - I do not understand the meaning.
L400 I’d change “3.7. In vitro differentiations” to “3.7. In vitro differentiation”
L402 “After positive result of differentiation” – Which differentiation?
L440 “expressed mRNAs gene expression” – not correct wording
L530-534 The confluence does not matter, initial cell plating density might be different, size of the cells might be different. Only number of population doubling could be compared in that context.
L551 USCs are NOT pluripotent, so “pluripotency” was NOT “consistent with currently accepted biological characteristics”
L565-567 “It is very important to preserve the viability of the few cells present in the samples, immediately after collection by addition of supplements such glucose, serum and other sources of nutrition, as PRP.” – That was not shown in the current study and thus shall not be in the Conclusion section.
Comments on the Quality of English LanguageQuality of English language shall be improved
Author Response
The article “Characterization of urine-derived stromal/stem cells from healthy dogs and dogs affected by chronic kidney disease (CKD)” describes properties of USCs derived from dogs using three experimental approaches.
The introduction section lacks important information about the species of the described cells. For each article, the source shall be presented: human, canine, murine (some other?).
ANSWER: the authors thank the referee for her/his suggestions and the requested information have been added
Figures 9 and 10 in the additional file are of bad quality.
ANSWER: the authors thank the referee for her/his suggestions. The authors have been trying to improve the images.
The idea that “It is very important to preserve the viability of the few cells present in the samples, immediately after collection by addition of supplements such glucose, serum and other sources of nutrition, as PRP.” was not shown in the current study and thus shall not be in the Conclusion section.
ANSWER: the authors thank the referee for her/his suggestions and the sentence has been modified
There are several minor issues listed below. Also extensive correction of English language is necessary.
Overall, I recommend minor revision of the MS.
L39 What is “immunogenic antigen”? Most antigens are immunogenic by definition.
ANSWER: the authors thank the referee for her/his suggestions and the sentence has been modified adding the specific antigen (major histocompatibility complex MHC II)
L56-L115 For each article, the source shall be presented: human, canine, murine (some other?)
ANSWER: the authors thank the referee for her/his suggestions, and we have added the different species
L113-114 “The results show that it is possible to isolate canine USCs from both healthy and CKD dogs, using a variety of urine collection methods.” - That is not true, only one method was successful, cells obtained by different methods were not tested.
ANSWER: the authors thank the referee for her/his suggestions, and the sentence has been modified
L174 “1% antibiotic/antimycotic” – which one?
ANSWER: the authors thank the referee for her/his suggestions, and we have specified the reagent
L249-251 Why two media are used, induction and maintenance? A reference to this protocol shall be included.
ANSWER: the authors thank the referee for her/his suggestions, but the reference has already been included and its number is 26: Romanov et al. 2003
L269 here and please check throughout the text: if RNA of DNA is analyzed, italics shall be used, if protein – not italics. Also “Oct4” is preferable here as capital letters are used only for human genes, not dogs.
ANSWER: the authors thank the referee for her/his suggestions, and we have checked all manuscript
L276 25 microliters?
ANSWER: the authors thank the referee for her/his suggestions and the mistake has been corrected
L281 “T1 GAPDH” – What is “T1”?
ANSWER: the authors thank the referee for her/his suggestions and the mistake has been corrected
L297 Celsius degree sign is wrong.
ANSWER: the authors thank the referee for her/his suggestions and the mistake has been corrected
L365 (Figure 3) and all graphs: why whiskers are looking both up only and up and down on one graph? Why on other graphs they are looking only up? Please use only one option on all the graphs. ANSWER: the authors thank the referee for her/his suggestions and the mistake has been corrected
L376 “At P5, these cells showed a slight minor proliferation compared to those at P1” - I do not understand the meaning.
ANSWER: the authors thank the referee for her/his comment. We meant that at P5, cells are older than P1 and the growth curve and number of cells obtained at each growth point is lower than those at P1.
L400 I’d change “3.7. In vitro differentiations” to “3.7. In vitro differentiation”
ANSWER: the authors thank the referee for her/his suggestions and the mistake has been corrected
L402 “After positive result of differentiation” – Which differentiation?
ANSWER: the authors thank the referee for her/his suggestions, and we have specified all three differentiations
L440 “expressed mRNAs gene expression” – not correct wording
ANSWER: the authors thank the referee for her/his suggestions and the mistake has been corrected
L530-534 The confluence does not matter, initial cell plating density might be different, size of the cells might be different. Only number of population doubling could be compared in that context.
ANSWER: the authors thank the referee for her/his comment but we have some difficulties understanding the issue. In any case, in the discussion, we have added a sentence regarding the ability of USCs to give rise in vitro to clones with frequency that increased with cell-seeding densities up to 1000 cells/cm2, indicating some paracrine signaling between urine derived cells, which may potentiate CFU formation during cell culture as reported in the paper of Sarugaser et al. (2005) and Lange-Consiglio et al. (2012).
L551 USCs are NOT pluripotent, so “pluripotency” was NOT “consistent with currently accepted biological characteristics”
ANSWER: the authors thank the referee for her/his comment, and we have corrected with multipotency
L565-567 “It is very important to preserve the viability of the few cells present in the samples, immediately after collection by addition of supplements such glucose, serum and other sources of nutrition, as PRP.” – That was not shown in the current study and thus shall not be in the Conclusion section.
ANSWER: the authors thank the referee for her/his comment and we have corrected removing the word “PRP”
Quality of English language shall be improved
ANSWER: the authors thank the referee for her/his comment, but the paper was proofread for English by a English mother tongue prior to submission but carefully revised now. This native speaker is a veterinary who was also co-editor of Small Animal Practice and vice-president of British Small Animal Veterinary Association.
Her name is
Katie McConnell (MA VetMB CertVR CertSAM MRCVS)
km.copyed@gmail.com
We feel that we have addressed all of the queries raised by the referees and hope that the paper is now acceptable for publication in Animals.
We thank you in advance for your time and consideration.
On behalf of all authors best regards,
Giulia Gaspari
Round 2
Reviewer 1 Report
Comments and Suggestions for Authors
The replies to the comments made and the changes to the manuscript are relevant to the request.